# Correlation and regression analysis between residual gradation and uncorrected visual acuity one year after refractive surgery with LASIK, FS-LASIK, PRK, PRK Xtra techniques and the implantation of ICL® posterior chamber phakic lens in myopic correction

**Irene Blanco-Dominguez**[1][�]*, **Francesc Duch**[2‡], **Vicente Polo**[3‡], **José M. Abad**[4‡], **Manuel Gomez-Barrera**[5,6‡], **Elena Garcia-Martin**[3‡]

1 Institut Clínic d'Oftalmologia (ICOF), Hospital Clínic, Barcelona, Spain, 2 Department of Refractive Surgery, Institut Català de la Retina, Barcelona, Spain, 3 Department of Ophthalmology, Miguel Servet University Hospital, Zaragoza, Spain, 4 Department of Public Health University of Zaragoza, Zaragoza, Spain, 5 San Jorge University, Zaragoza, Spain, 6 Pharmacoeconomics & Outcomes Research Iberia (PORIB), Madrid, Spain

☺ These authors contributed equally to this work.
‡ FD, VP, JMA, MGB and EGM also contributed equally to this work.
* ire_blanco@hotmail.com

## Abstract

### Purpose

To analyze the influence of the final spherical equivalent (SE) in LogMAR uncorrected distance visual acuity (UDVA) one year after refractive surgery. We analysed refractive results, their predictability and efficacy, and the safety results of the different methods as secondary outcomes.

### Setting

Refractive Surgery Unit of the Institut Català de Retina (ICR) in Barcelona, Spain.

### Design

Retrospective, analytical observational study.

### Methods

Retrospective and observational study of 654 eyes of 327 patients who underwent refractive surgery to treat their myopia or myopic astigmatism using LASIK, FS-LASIK, PRK, PRK Xtra or ICL-type lens implantation surgery were included.

### Results

The correlation between the SE in absolute value was statistically significant in all techniques utilized, reaching higher values in the FS-LASIK and LASIK techniques, 0.774 and

**Data Availability Statement:** All relevant data are within the manuscript and its Supporting Information files.

**Funding:** The authors received no specific funding for this work.

**Competing interests:** The authors have declared that no competing interests exist.

0.706 respectively, and lesser values in PRK (0.480) and PRK Xtra (0.482). A significant adjustment via a univariate linear regression model could be implemented in all techniques, albeit the $R^2$ coefficient of determination values were higher than those for the FS-LASIK (0.599) and LASIK (0.494) techniques.

## Conclusions

There is a positive correlation between post-surgical SE value and post-operative LogMAR UDVA. These regression models can be adjusted to predict the final UDVA according to the final SE. The techniques that are most influenced by the final SE in terms of their visual results are FS-LASIK and LASIK.

## Introduction

Gradation, or residual refraction, is the most common complication of any refractive procedure. It encompasses overcorrection, under-correction, induced or residual astigmatism, and regression [1–3]. Some patients may adapt to small degrees of residual gradation, and for this reason, the main indicator to consider retreatment in these cases is patients' dissatisfaction with visual acuity (VA). This is why retreatment rate does not assess surgical precision, which should be measured by the percentage of patients achieving emmetropia.

It has been observed that regression generally progresses for the first 3–6 months after Laser-Assisted In Situ Keratomileusis (LASIK) [4] and that retreatment must be performed after refraction has stabilized [4–6]. Reinitiating treatment entails the risk of inducing corneal ectasia in cases of stromal ablation treatment (retreatment or bioptics), 250–300 μm being the lowest accepted limit for residual corneal bed thickness [7,8]. Likewise, the exchange in posterior chamber phakic lens is not less resistant to complications.

While it is taken for granted that residual gradation directly affects the visual result of surgical techniques, we have not found any study in the scientific literature that directly evaluates and compares either this factor or its repercussions across the varied techniques. The main objective of this study is to analyze the influence of the final spherical equivalent (SE) in LogMAR uncorrected distance VA (UDVA) in each of the following refractive techniques independently after one year: LASIK with flap creation using mechanical microkeratome or femtosecond laser (FS-LASIK), Photorefractive keratectomy (PRK), PRK with concurrent crosslinking (PRK Xtra) and Implantable Collamer Lens (ICL)-type posterior chamber intraocular phakic lens (Staar Collamer Implantable Contact Lens, STAAR Surgical AG). We analyzed refractive results, their predictability and efficacy, and the safety results of the different methods as secondary outcomes.

## Material and methods

This project was designed as a retrospective, analytical observational study. The study population are patients who underwent refractive surgery in 2015 (January-December) at the refractive surgery ward of the Institut Català de Retina (ICR) in Barcelona (Spain), and who completed a year of monitoring. The main information source was the center's electronic medical history database.

Inclusion criteria included: ≥21 years of age; myopic astigmatism or myopic vision with or without associated astigmatism; patients who had undergone LASIK, FS-LASIK, PRK, PRK

Xtra or ICL[®]-type lens implantation surgery with no contraindications against the selected technique; binocular surgery with the same surgical technique; surgical objective of emmetropia and stable gradation during the last 12 months prior to surgery (refractive change less than 0.5 D). The exclusion criteria were as follows: ocular or systemic pathology counter-indicating refractive surgery, a history of prior surgery, suspicious keratoconus topography, associated presbyopia and amblyopia (corrected distance VA (CDVA) $\leq$ 20/32 Snellen).

This study has been approved by the ICR research committee (ICR-21/17) and was carried out according to the principles and basic ethical regulations originated in the Declaration of Helsinki (Fortaleza, Brazil version; October 2013) approved by the World Medical Association. All the patients included in the study gave their written informed consent to carry out the interventions and manage their data for scientific purposes.

The variables included within the study are surgical technique, age, sex, pre-surgery sphere, cylinder and SE, pre-surgery LogMAR UDVA, pre-surgery LogMAR CDVA, efficacy index (post-operative UDVA/ pre-operative CDVA), safety index (post-operative CDVA/ pre-operative CDVA), final sphere, cylinder and SE at one year (in absolute value), percentage of cases with final SE between ±0.25 D, 0.5 D and 0.75 D, and LogMAR UDVA at one year.

In all cases of corneal surgery, corneal ablation was performed using the Alcon[®] Wave-Light[®] EX500 excimer laser with an aspheric ablation (wavefront optimized) profile. To create the corneal flap using the LASIK technique, the Bausch & Lomb[®] Zyoptix XP[®] microkeratome was used, and for the FS-LASIK technique, the Alcon[®] Wavelight FS200 Femtosecond Laser[®] was used. In PRK, epithelial removal was performed manually with a spatula after applying a 20% alcohol solution for 20 seconds. In cases of ablations measuring more than 70 microns, mitomycin C was added intraoperatively at 0.02% for 12 seconds so as to prevent corneal haze. In PRK Xtra, accelerated crosslinking was performed after excimer treatment, using riboflavin 0.1% (MedioCROSS M[®], Avedro) and the CCL Vario Illumination System[®] (Ophtec[®], AJ Groningen Schweitzerlaan) following the Peschkel L protocol with a total energy level of 2.70 J/cm$^2$, for two minutes in continuous mode, with an ultraviolet energy of 18mW/cm$^2$. All ICL[®] lenses implanted were model V4c lenses, which incorporate central Aquaport; iridectomy is therefore unnecessary. For cases of $\geq$1.5 D refractive astigmatism, the toric lens model was implanted. In cases of astigmatism between 0.5 and 1.25 D, the main incision of 3 mm was made in the corneal meridian with the steepest curve, or limbal relaxing incisions were associated as per the Donnenfeld nomogram [9].

## Statistical analysis performed

Frequencies and percentages of the average and qualitative variables, standard deviation (SD), and range in quantitative variables have been calculated for the descriptive analysis of patients. The presence of significant differences between groups was analyzed using the chi-squared test for qualitative variables, and the ANOVA for quantitative variables. The homogeneity of the variances was verified. The post-hoc Scheffé test and the Games-Howell procedure were carried out for the cases of homogeneity and heterogeneity respectively.

To evaluate the relationship between the final SE and the LogMAR UDVA at one year in the various techniques utilized, a linear correlation analysis was completed using the Pearson r correlation and a simple linear regression model, evaluated using the $R^2$ coefficient of determination and the model's significance level. Absolute value was taken for the SE to facilitate interpretation of results.

The data were processed with SPSS software, whose license for use belongs to the University of Zaragoza, with a threshold value of $p = 0.05$ set for the acceptance or rejection of null hypotheses.

## Results

Data from a total of 654 eyes were analyzed, corresponding to 327 patients: 133 men (40.7%) and 194 women (59.3%). The average age (±SD) was 31.8 (±5.4) years of age with a range (43–20). The average SE (±SD) prior to treatment was -4.11 (±2.33) with a range (-0.50; -18.875). The average (±SD) LogMAR UDVA was 1.125 (±0.272) with a range (1.301 - 0), and the average (±SD) LogMAR CDVA prior to treatment was -0.036 (±0.047) with a range (0.155 - -0.079). The values of the variables across the groups are presented in Table 1.

Regarding demographic variables, statistically significant differences were observed in terms of gender distribution, wherein a smaller percentage of women underwent PRK interventions (Pearson chi-squared test, p<0.001) and patients who underwent LASIK surgery were younger than those with ICL (Scheffé, p = 0.015).

With regard to the SE prior to surgery, only those patients who underwent LASIK and FS-LASIK presented comparable values. In the rest of the comparisons, objective values of p≤0.002 were set with the Games-Howell procedure, indicating fewer negative values in PRK and PRK Xtra, and more negative values in ICL. Only groups of LASIK and FS-LASIK showed comparable values of sphere (p>0.05), whereas ICL group had a higher relevant pre-surgical cylinder (p<0.05).

In LogMAR UDVA prior to surgery, only LASIK, FS-LASIK and PRK Xtra produced values comparable between them. Statistically significant differences were found using the Games-Howell test in the rest of the techniques: PRK vs. PRK Xtra p = 0.013, ICL vs. PRK Xtra p = 0.001, and p<0.001 in the rest of the comparisons, with the highest values found (worst prior to uncorrected vision) in ICL and the lowest in PRK.

In LogMAR CDVA prior to surgery, ICL had statistically higher values (worst corrected vision) than the rest, with values of p = 0.012 compared to PRK Xtra and p<0.001 in all other cases, all of them calculated using the Scheffé test.

The correlation between the SE in absolute value and LogMAR UDVA was statistically significant in all techniques utilized, reaching higher values in the FS-LASIK and LASIK techniques, 0.774 and 0.706 respectively, and lower values in PRK (0.480) and PRK Xtra (0.482). Table 2 presents the SE Pearson correlation coefficient values in terms of absolute value, contrasted with LogMAR UDVA at one year after surgery, as well as its statistical significance with

**Table 1. Results of demographic and pre-intervention variables by techniques.**

| Variable | FS-LASIK n = 216 | LASIK n = 114 | PRK n = 180 | PRK Xtra n = 32 | ICL n = 112 | p Value ANOVA |
|---|---|---|---|---|---|---|
| Women, n(%) | 136 (63.0) | 76 (66.7) | 82 (45.6) | 20 (62.5) | 74 (66.1) | <0.001 |
| Average age±SD (range) | 32.0±5.8 (42–20) | 30.1±5.0 (42–23) | 32.0±5.3 (43–21) | 31.8±5.3 (39–24) | 32.7±5.0 (43–24) | 0.007 |
| Average SE±SD (range) | -4.091±1.602 (-0.750; -7.875) | -3.775±1.374 (-0.875; -7.625) | -2.391±0.871 (-0.50; -5.75) | -3.039±0.784 (-1.50; -4.75) | -7.566±2.547 (-3.00; -18.875) | <0.001 |
| Average sphere±SD (range) | -3.721 ±1.663 (0; -7.75) | -3.458 ±1.394 (0; -7.00) | -2.075 ±0.881(-0.50; -5.50) | -2.750 ±0.847 (-1.25; -4.50) | -7.069 ±2.572 (-2.50; -18.50) | <0.001 |
| Average cylinder±SD (range) | 0.741±0.676 (0;3.75) | 0.634±0.588 (0;3.25) | 0.632±0.474 (0;3.50) | 0.578±0.437 (0;1.50) | 0.993±0.848 (0;4.50) | <0.001 |
| Average UDVA±SD (range) | 1.171±0.242 (1.301; 0.301) | 1.173±0.225 (1.301; 0.398) | 0.942±0.318 (1.301; 0) | 1.103±0.237 (1.301; 0.398) | 1.293±0.063 (1.301; 0.699) | <0.001 |
| Average CDVA±SD (range) | -0.038±0.044 (0.097; -0.079) | -0.044±0.041 (0.046; -0.079) | -0.047±0.041 (0.097; -0.079) | -0.043±0.046 (0.046; -0.079) | -0.010±0.059 (0.155; -0.079) | <0.001 |

SE: spherical equivalent; SD: standard deviation; UDVA: uncorrected distance visual acuity; CDVA: corrected distance visual acuity.

**Table 2. Correlation results and regression models.**

| Technique | Correlation SE/UDVA LogMAR, (p) | Regression model | ANOVA regression, p Value | R2 |
|---|---|---|---|---|
| FS-LASIK | 0.774 (<0.001) | UDVA LogMar = -0.076+0.313SE | <0.001 | 0.599 |
| LASIK | 0.706 (<0.001) | UDVA LogMar = -0.056+0.243SE | <0.001 | 0.494 |
| PRK | 0.480 (<0.001) | UDVA LogMar = -0.062+0.191SE | <0.001 | 0.226 |
| PRK Xtra | 0.482 (<0.001) | UDVA LogMar = -0.020+0.126SE | 0.005 | 0.207 |
| ICL | 0.603(<0.001) | UDVA LogMar = -0.044+0.207SE | <0.001 | 0.357 |

SE: spherical equivalent; UDVA: uncorrected distance visual acuity.

a contrast value of zero. The various regression models are likewise displayed along with the ANOVA and the $R^2$ coefficient of determination value of the model.

A second correlation analysis was performed in the FS-LASIK group to see if myopia prior to surgery influenced the correlation results. Myopia was divided into mild ($\leq$ 3D; 78 eyes; 36.1%), moderate (from 3 to 6 D; 114 eyes; 52.8%) and high ($\geq$ 6D; 24 eyes, 11.1%). The correlation measured with the Pearson coefficient between post-surgical SE value and post-operative LogMAR UDVA in the FS-LASIK mild myopia group was 0.771 (p <0.001), 0.785 (p <0.001) in the FS-LASIK moderate myopia group, and 0.86 (p <0.001) in the FS-LASIK group with high myopia. In the ICL group, two groups were formed according to myopia before the intervention, with myopia being divided into mild-moderate ($\leq$ 6D; 36 eyes, 32.14%) and high ($\geq$ 6D; 76 eyes, 67.9%). The correlation measured with the Pearson coefficient between post-surgical SE value and post-operative LogMAR UDVA in the mild-moderate myopia ICL group was 0.624 (p <0.001) and 0.599 (p <0.001) in the high myopia ICL group. Very similar results were obtained in all subgroups regardless of initial myopia.

A significant adjustment via a univariate linear regression model could be implemented in all techniques, albeit the $R^2$ coefficient of determination values were higher than those for the FS-LASIK (0.599) and LASIK (0.494) techniques. The regression graphics for the different techniques are shown in Fig 1.

Absolute value of SE, sphere, cylinder and LogMAR UDVA can be observed in Table 3 in the different techniques per year.

The predictability results of each method, defined as the percentage of cases in which the final SE is between ±0.50 D, are included in Table 4. In the ICL group, 70.5% of lenses were VICMO models and 29.5% were VTICMO toric lens models.

The results of the comparison between the objective or intended correction (pre-surgical SE value) with the correction eventually achieved (which is obtained by subtracting the final SE value from the pre-surgical SE) of each technique are shown in the data spread graphic in Fig 2.

Results of effectiveness and safety can be seen in Fig 3. A statistically significant improvement in CDVA was observed in the ICL group (p<0001).

## Discussion

The indications for each refractive technique are different, as a result groups cannot be compared. The effect of residual graduation was analyzed independently for each surgical technique. We detected correspondence in all the refractive techniques assessed between post-surgical SE and post-operative UDVA, both as measured one year after surgery. This correlation is positive; the absolute value of the final SE increases as the value of the LogMAR UDVA increases, meaning worse vision is produced, and it seems not to be influenced by previous myopia. Based on this relationship, regression models can be adjusted in order to predict the

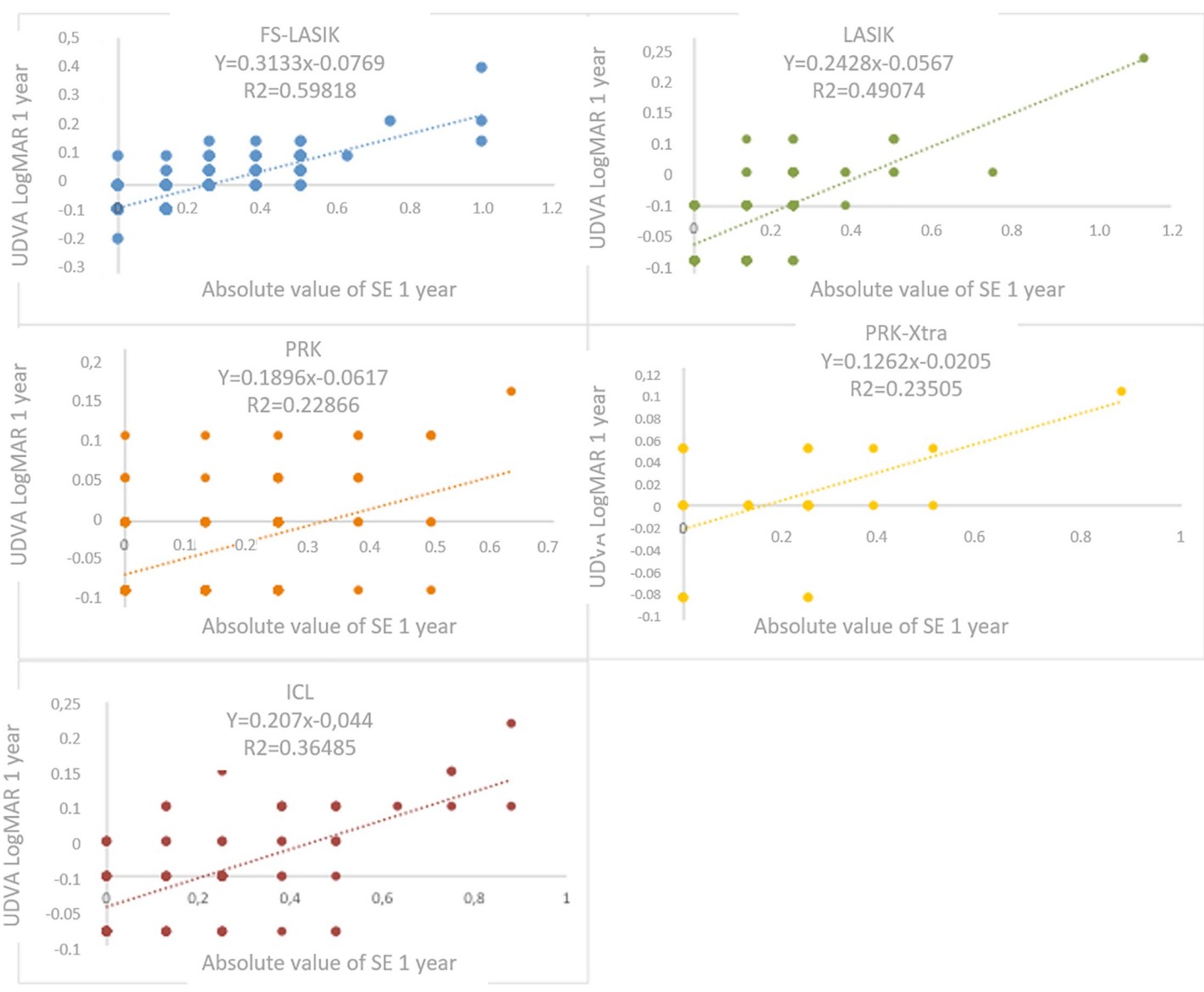

**Fig 1. Regression graphs of the different techniques.**

final UDVA according to the final SE with an average $R^2$. This demonstrates that these are multifactor techniques, which are influenced by other variables that affect the final UDVA and must be included in later research. In this respect, the techniques that are most influenced by

**Table 3. Descriptive results of absolute value of SE, sphere, cylinder and UDVA LogMAR per year in the different techniques.**

| | FS-LASIK | LASIK | PRK | PRK Xtra | ICL |
|---|---|---|---|---|---|
| Average SE ±SD | 0.19±0.19 | 0.18±0.17 | 0.14±0.14 | 0.20±0.20 | 0.21±0.21 |
| Average sphere ±SD (range) | 0.08±0.17 (-1;0,50) | 0.10±0.18 (-1;0.75) | 0.05±0.17 (-0.50;0.75) | 0.19±0.31 (0;1.25) | 0.17±0.22 (-0.50;0.75) |
| Average cylinder ±SD (range) | 0.30±0.26 (0;1) | 0.32±0.25 (0;1) | 0.26±0.25 (0;1.25) | 0.44±0.23 (0;0.75) | 0,33±0.29 (0;1.25) |
| Average UDVA ±SD | -0.0164±0.078 | -0.0118±0.06 | -0.0349±0.055 | 0.0053±0.052 | -0.0003±0.073 |

SE: spherical equivalent; SD: standard deviation; UDVA: uncorrected distance visual acuity.

**Table 4. Predictability of refractive results: Percentage of cases with final spherical equivalent (SE) between ± 0.25, ± 0.50 and ± 0.75 diopters.**

| Range (SE) | FS-LASIK (n = 216) | LASIK (n = 114) | PRK (n = 180) | PRK Xtra (n = 32) | ICL (n = 112) |
|---|---|---|---|---|---|
| ±0.25,% (CI95%) | 82.0 (76.9–87.1) | 90.3 (84.9–95.7) | 91.7 (87.7–95.7) | 78.2 (63.9–92.5) | 75.9 (68.0–83.8) |
| ±0.50,% (CI95%) | 97.3 (95.1–99.5) | 97.4 (94.5–100.0) | 99.4 (98.3–100.0) | 97 (91.1–100.0) | 94.7 (90.6–98.9) |
| ±0.75,% (CI95%) | 98.7 (97.2–100.0) | 99.2 (97.6–100.0) | 100% | 97 (91.1–100.0) | 98.3 (95.9–100.0) |

CI: confidence interval.

the final SE in uncorrected vision are FS-LASIK ($R^2$ = 0.599) and LASIK ($R^2$ = 0.494). In clinical practice, this means that FS-LASIK can be predicted by the final SE with nearly 60% of UDVA variability.

Routine PRK procedures are usually performed in our center when myopia is equal to or less than 2 diopters and in some specific cases, without exceeding 5–6 diopters of myopia or cylinders over 4 diopters. Mitomycin C 0.02% must be added when the ablation surpasses 70 microns. PRK Xtra is usually reserved for cases of irregular topographies without keratoconus indices, with a pachymetry limit of 400 microns without epithelium. Treatment with FS-LA-SIK / LASIK is performed up to a maximum of 8 diopters of myopia and 6 of cylinder, tending increasingly to perform all femtosecond laser treatments for the creation of the flap. Phakic intraocular lens surgery is generally considered from 6–7 diopters of myopia and in cases in which laser treatment is not viable.

Taking only into consideration our descriptive results and keeping in mind that groups cannot be compared between themselves, patients who underwent PRK started from lower SEs and obtained higher UDVA's as well as the best SE and predictability but showed the lowest $R^2$ after PRK Xtra in the regression analysis. This could suggest that more aspects affect this technique. It is logical to think that surface ablation techniques such as PRK and PRK Xtra may be influenced by more factors than the rest, since a more extensive healing process takes place. An inappropriate healing response with large quantities of active keratinocytes and an overly-large extracellular matrix produce subepithelial haze. The occurrence of some haze is common, but only those patients with pathological scarring develop clinically relevant haze. Haze intensity is greater during the first 6 months after PRK, tending to decrease over the following 12–24 months [10]. The development of haze may be modified with the use of steroids during the post-operative period and may be prevented by the intraoperative use of topical mitomycin as a prophylactic when ablation exceeds -4 to -6 D [11] or if specific ablation depth exceeds 50–75 μm [12]. In our series, mitomycin C at 0.02% was used in 14 eyes (7.8%), and 12 eyes (6.8%) presented diffuse haze at one month after the surgery. This disappeared in all cases at one year. In the case of PRK Xtra, opacification similar to PRK corneal haze may occur, but it is much deeper and is generally paracentrally located due to associated cross-linking. It is compatible with good visual results, but we must consider that it is another added aspect that can alter the corneal healing.

The current trend is to be even more conservative with corneal tissue, and for this reason, correction with posterior camera phakic lenses is rated beyond -6 D. This is mostly because ablation must be performed on a large quantity of tissue in high myopias, incurring problems such as regression or haze in the case of PRK [13,14], or compromising the long-term biomechanical stability of the cornea in LASIK or FS-LASIK [7,15–17]. It is important to remember that the ICL group is a large group of myopic patients whose starting vision is more limited and who may be influenced by more factors when results are interpreted. The lower predictability associated with this technique could also be explained by a series of limitations, such as the fact that lens powers vary by grades of -0.5 D, and that the management of the

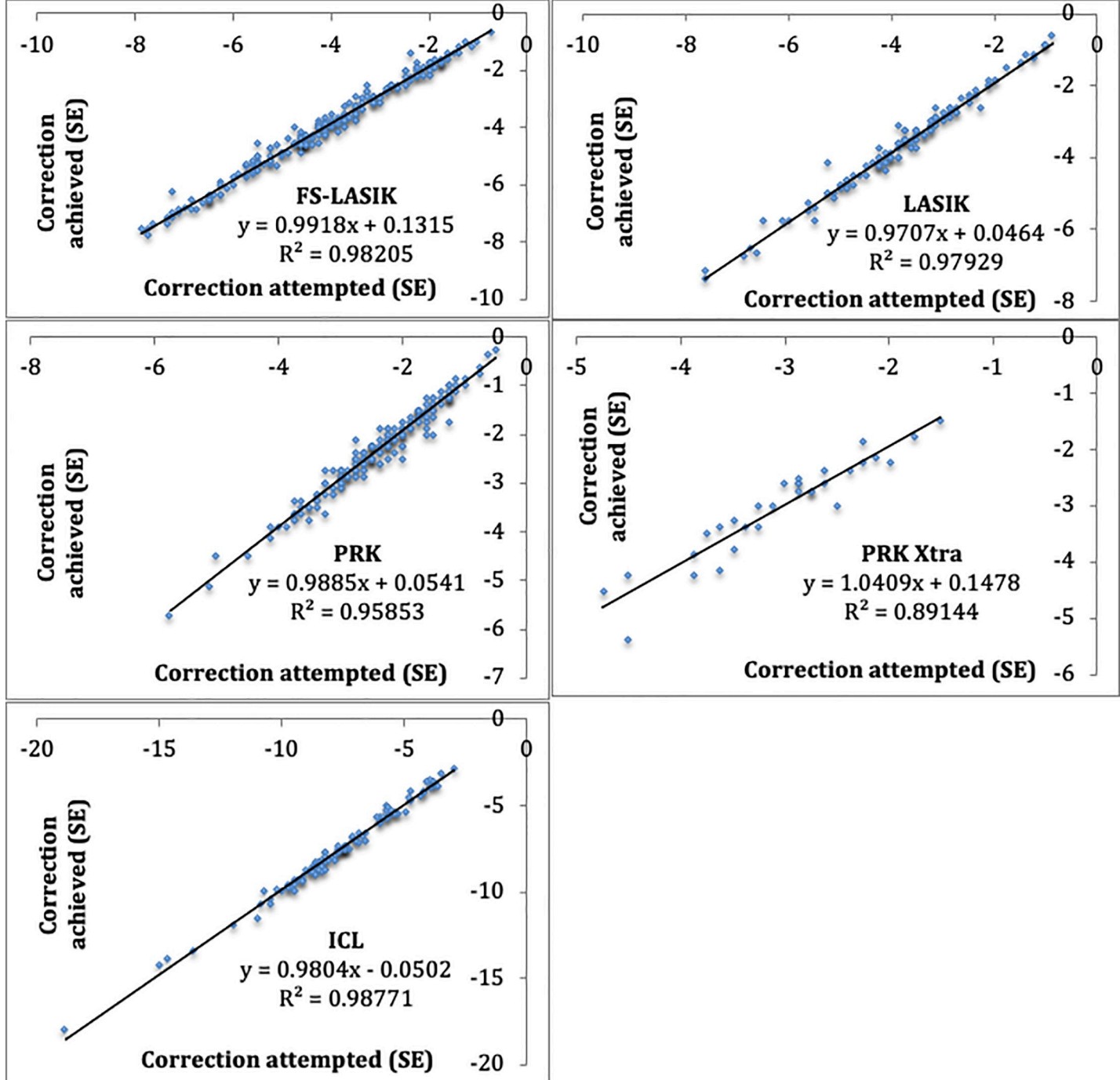

**Fig 2. Relation of the correction attempted VS the correction achieved (SE).**

corneal cylinder of less than 1.5 D is performed through incisions, in our case, which entails more variability. Even after taking these factors into account, the adjustment in these patients' corrections was almost perfect, as is shown in Fig 2, with an $R^2$ of 0.99, and postsurgical cylinder was similar to other methods (Table 3).

Despite a low quality of evidence, it was observed that the visual and refractive results of LASIK and PRK in the treatment of myopia are similar [18]. It was also suggested in the

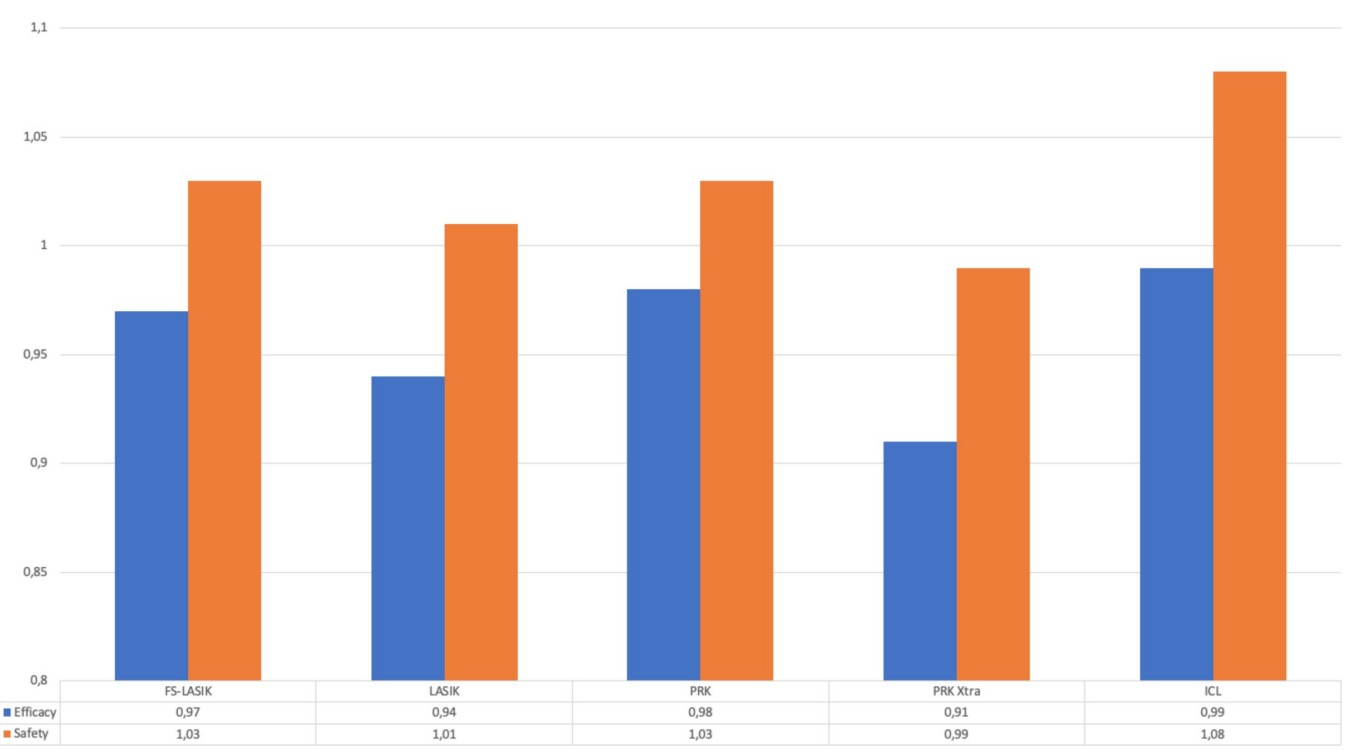

| | FS-LASIK | LASIK | PRK | PRK Xtra | ICL |
|---|---|---|---|---|---|
| ■ Efficacy | 0,97 | 0,94 | 0,98 | 0,91 | 0,99 |
| ■ Safety | 1,03 | 1,01 | 1,03 | 0,99 | 1,08 |

**Fig 3. Results of the efficacy and safety indices of the different techniques.**

Cochrane's review [19] that one year after surgery the phakic lenses are safer than the excimer laser for the surgical correction of moderate-high myopia in the range of -6 to -20 diopters. Although our groups cannot be compared, visual and refractive outcomes of each technique prove the current evidence.

Lastly, it is important to stress the limitations of this study. This is a retrospective study of a series of cases at a single center, in which hypermetropic patients were excluded. For upcoming projects in this subject, the ideal design would be a prospective, analytical interventional study to validate conclusions reached and collected in this study.

## Conclusions

Graduation or residual refraction is the most common condition of any refractive procedure. Although it is assumed that residual graduation directly affects the visual outcome of the different surgical techniques, in clinical practice it appears that some techniques are more sensitive to refractive defects than others. This is the first study that evaluates this aforementioned factor and its impact on different methods, showing a positive correlation between post-surgical SE value and post-operative LogMAR UDVA. These regression models can be adapted to predict the final UDVA according to the final SE, and the techniques that are most influenced by the final SE in terms of their visual results are FS-LASIK and LASIK. Further research is needed to incorporate more variables in the equation to achieve higher values of $R^2$ than the ones obtained in the present study.

## Supporting information

**S1 Data.**
(XLSX)

## Author Contributions

**Conceptualization:** Irene Blanco-Dominguez, Francesc Duch, Vicente Polo.

**Data curation:** Irene Blanco-Dominguez, Manuel Gomez-Barrera.

**Formal analysis:** Irene Blanco-Dominguez, José M. Abad, Manuel Gomez-Barrera.

**Investigation:** Irene Blanco-Dominguez.

**Methodology:** Irene Blanco-Dominguez, Francesc Duch, José M. Abad.

**Project administration:** Irene Blanco-Dominguez, Francesc Duch, Vicente Polo.

**Resources:** Irene Blanco-Dominguez.

**Software:** Irene Blanco-Dominguez, Manuel Gomez-Barrera.

**Supervision:** Irene Blanco-Dominguez, Francesc Duch, Vicente Polo, José M. Abad, Manuel Gomez-Barrera, Elena Garcia-Martin.

**Validation:** Irene Blanco-Dominguez, Francesc Duch, Vicente Polo, José M. Abad, Manuel Gomez-Barrera, Elena Garcia-Martin.

**Visualization:** Irene Blanco-Dominguez, Francesc Duch, Elena Garcia-Martin.

**Writing – original draft:** Irene Blanco-Dominguez.

**Writing – review & editing:** Irene Blanco-Dominguez, Francesc Duch, Vicente Polo, José M. Abad, Manuel Gomez-Barrera, Elena Garcia-Martin.

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
