## [Decision Letter · Decision Letter 0]

13 Jul 2020

PONE-D-20-08468

Correlation and regression analysis between residual gradation and uncorrected visual acuity one year after refractive surgery with LASIK, FS-LASIK, PRK, PRK Xtra techniques and the implantation of ICL® posterior chamber phakic lens.

PLOS ONE

Dear Dr. %Blanco-Dominguez%,

Thank you for submitting your manuscript to PLOS ONE. After careful consideration, we feel that it has merit but does not fully meet PLOS ONE’s publication criteria as it currently stands. Therefore, we invite you to submit a revised version of the manuscript that addresses the points raised during the review process.

We look forward to receiving your revised manuscript.

Kind regards,

Rajiv R. Mohan, Ph.D.

Academic Editor

PLOS ONE

Journal Requirements:

3. Your ethics statement must appear in the Methods section of your manuscript. If your ethics statement is written in any section besides the Methods, please move it to the Methods section and delete it from any other section. Please also ensure that your ethics statement is included in your manuscript, as the ethics section of your online submission will not be published alongside your manuscript.

Reviewers' comments:

Reviewer's Responses to Questions

**Comments to the Author**

1. Is the manuscript technically sound, and do the data support the conclusions?

Reviewer #1: Partly

Reviewer #2: Yes

2. Has the statistical analysis been performed appropriately and rigorously? 

Reviewer #1: Yes

Reviewer #2: Yes

3. Have the authors made all data underlying the findings in their manuscript fully available?

Reviewer #1: Yes

Reviewer #2: Yes

4. Is the manuscript presented in an intelligible fashion and written in standard English?

Reviewer #1: Yes

Reviewer #2: Yes

5. Review Comments to the Author

Reviewer #1: This paper addresses an important and interesting problem- correlation analysis between residual gradation and UDVA after different refractive surgery methods. But the paper needs very significant improvement before acceptance for publication. My detailed comments are as follows:

1.There was a statistically significant difference in SE in each group before surgery. The correlation between postoperative SE and LogMAR UDVA is closely related to myopic refractive power before surgery, not the refractive surgery techniques. Therefore, it is recommended to add the correlation statistics of postoperative SE and LogMAR UDVA in high, medium and low diopters in each group. At least, choose one of the surgical methods to statistically analyze the correlation between postoperative SE and LogMAR UDVA according to the preoperative SE.

2.The second paragraph of the introduction mainly discusses the cause of retreatment, the choice of retreatment and the risk of retreatment, which has not much relevance to the subject of the article and can be reduced.

Reviewer #2: For upcoming project in this subject, I would recommend a design of the study as a prospective, analytical interventional study to validate the conclusion of this one

You should include the word myopic/ myopic correction in the main title as you exclude hyperopic or pure astigmatic corrections

It is not clear which protocol you use for PRK Xtra, you should specify that Peschke protocol consist of: a total energy level of 2.70 J/cm2 for “X” seconds using which kind of riboflavin ( iso/hypotonic, dextran free?, etc). As you set that postoperative haze was peripheral, could you explain the cause of this observation?.

6. PLOS authors have the option to publish the peer review history of their article (what does this mean?). If published, this will include your full peer review and any attached files.

Reviewer #1: No

Reviewer #2: No

---

## [Author Response · Author response to Decision Letter 0]

23 Jul 2020

Reviewer #1: 

1.There was a statistically significant difference in SE in each group before surgery. The correlation between postoperative SE and LogMAR UDVA is closely related to myopic refractive power before surgery, not the refractive surgery techniques. Therefore, it is recommended to add the correlation statistics of postoperative SE and LogMAR UDVA in high, medium and low diopters in each group. At least, choose one of the surgical methods to statistically analyze the correlation between postoperative SE and LogMAR UDVA according to the preoperative SE.

The reviewer makes an interesting observation in this regard, however we can consider how there is no greater correlation depending on the initial graduation results, since techniques such as PRK and PRK xtra that start from lower graduations, have a lower correlation between post-surgical SE value and post-operative LogMAR UDVA than LASIK or FS-LASIK starting from higher graduations. A correlation subanalysis has been carried out according to the initial myopia in the FS-LASIK and ICL technique, seeing that the results are very similar regardless of the previous graduation from which it was started, it is more, in FS-LASIK even slightly better the older previous graduations, reversing this fact with ICL, so it is the chosen surgical technique that gives rise to the affected values.

“A second correlation analysis was performed in the FS-LASIK group to see if myopia prior to surgery influenced the correlation results. Myopia was divided into mild (≤ 3D; 78 eyes; 36.1%), moderate (from 3 to 6 D; 114 eyes; 52.8%) and high (≥ 6D; 24 eyes, 11.1%). The correlation measured with the Pearson coefficient between post-surgical SE value and post-operative LogMAR UDVA in the FS-LASIK mild myopia group was 0.771 (p <0.001), 0.785 (p <0.001) in the FS-LASIK moderate myopia group, and 0.86 (p <0.001) in the FS-LASIK group with high myopia. In the ICL group, two groups were formed according to myopia before the intervention, with myopia being divided into mild-moderate (≤ 6D; 36 eyes, 32.14%) and high (≥ 6D; 76 eyes, 67.9%). The correlation measured with the Pearson coefficient between post-surgical SE value and post-operative LogMAR UDVA in the mild-moderate myopia ICL group was 0.624 (p <0.001) and 0.599 (p <0.001) in the high myopia ICL group. Very similar results were obtained in all subgroups regardless of initial myopia.”

2.The second paragraph of the introduction mainly discusses the cause of retreatment, the choice of retreatment and the risk of retreatment, which has not much relevance to the subject of the article and can be reduced.

The second paragraph has been reduced: “It has been observed that regression generally progresses for the first 3-6 months after Laser-Assisted In Situ Keratomileusis (LASIK) 4 and that retreatment must be performed after refraction has stabilized 4-6. Reinitiating treatment entails the risk of inducing corneal ectasia in cases of stromal ablation treatment (retreatment or bioptics), 250-300 µm being the lowest accepted limit for residual corneal bed thickness 7,8. Likewise, the exchange in posterior chamber phakic lens is not less resistant to complications.”

Reviewer #2: 

1.For upcoming project in this subject, I would recommend a design of the study as a prospective, analytical interventional study to validate the conclusion of this one.

I agree. The following sentence has been added in the discussion: “For upcoming projects in this subject, the ideal design would be a prospective, analytical interventional study to validate conclusions reached and collected in this study.

2. You should include the word myopic/ myopic correction in the main title as you exclude hyperopic or pure astigmatic corrections.

The tittle has changed: “Correlation and regression analysis between residual gradation and uncorrected visual acuity one year after refractive surgery with LASIK, FS-LASIK, PRK, PRK Xtra techniques and the implantation of ICL® posterior chamber phakic lens in myopic correction.”

3.It is not clear which protocol you use for PRK Xtra, you should specify that Peschke protocol consist of: a total energy level of 2.70 J/cm2 for “X” seconds using which kind of riboflavin ( iso/hypotonic, dextran free?, etc). 

The specifications have been made: “In PRK Xtra, accelerated crosslinking was performed after excimer treatment, using riboflavin 0.1% (MedioCROSS M®, Avedro) and the CCL Vario Illumination System® (Ophtec®, AJ Groningen Schweitzerlaan) following the Peschkel L protocol with a total energy level of 2.70 J/cm2, for two minutes in continuous mode, with an ultraviolet energy of 18mW/cm2.”

---

## [Editor Report · Decision Letter 1]

17 Aug 2020

Correlation and regression analysis between residual gradation and uncorrected visual acuity one year after refractive surgery with LASIK, FS-LASIK, PRK, PRK Xtra and ICL® techniques in myopic correction.

PONE-D-20-08468R1

Dear Dr. Blanco-Dominguez,

We’re pleased to inform you that your manuscript has been judged scientifically suitable for publication and will be formally accepted for publication once it meets all outstanding technical requirements.

Kind regards,

Rajiv R. Mohan, Ph.D.

Academic Editor

PLOS ONE
---

## [Editor Report · Acceptance letter]

24 Aug 2020

PONE-D-20-08468R1 

Correlation and regression analysis between residual gradation and uncorrected visual acuity one year after refractive surgery with LASIK, FS-LASIK, PRK, PRK Xtra and ICL® techniques in myopic correction. 

Dear Dr. Blanco-Dominguez:

I'm pleased to inform you that your manuscript has been deemed suitable for publication in PLOS ONE. Congratulations! Your manuscript is now with our production department. 

Kind regards, 

on behalf of

Dr. Rajiv R. Mohan 

Academic Editor

PLOS ONE